# TalkPlayData 2: An Agentic Synthetic Data Pipeline for Multimodal Conversational Music Recommendation

## Abstract

We present TalkPlayData 2, a synthetic dataset for multimodal conversational music recommendation generated by an agentic data pipeline. In the proposed pipeline, multiple large language model (LLM) agents are created under various roles with specialized prompts and access to different parts of information, and the chat data is acquired by logging the conversation between the Listener LLM and the Recsys LLM. To cover various conversation scenarios, for each conversation, the Listener LLM is conditioned on a finetuned conversation goal. Finally, all the LLMs are multimodal with audio and images, allowing a simulation of multimodal recommendation and conversation. In the LLM-as-a-judge and subjective evaluation experiments, TalkPlayData 2 achieved the proposed goal in various aspects related to training a generative recommendation model for music.[1]

## 1 Introduction

Conversational recommendation systems provide recommendations through a natural language dialog with users, requiring both multi-turn recommendation capabilities and natural language response generation (Goker & Thompson, 2000; Christakopoulou et al., 2016; Zhang et al., 2018). At each turn, these systems predict ranked lists of relevant music tracks based on conversation history and user queries while understanding preferences, context, and diverse query types. Beyond recommendation, systems generate engaging responses that describe recommendations and assist users in music exploration through natural language explanations (Doh et al., 2024). For example, in the music domain, early approaches leveraged dense embeddings to find appropriate music by computing similarity between multi-turn history embeddings and item embeddings (Chaganty et al., 2023; Doh et al., 2024; Melchiorre et al., 2025), although these methods suffered from architectural limitations on generating natural responses.

The main blockers for developing such a system may have been the lack of large-scale and high-quality datasets. For example, CPCD, a human-curated dataset, consists of only 917 conversations in total (Chaganty et al., 2023). Recent studies, such as Talk The Walk, LP-MusicDialog, and TalkPlay (Leszczynski et al., 2023; Doh et al., 2024; 2025), have been proposed to address this issue by actively adopting language models. Essentially, those methods consists of two stages: determining a music sequence and generating corresponding utterances. In (Leszczynski et al., 2023), the music sequence is determined based on several similarity assumptions, while in (Doh et al., 2024; 2025), it is determined by cascaded attribute filtering among a pool of music (a playlist). Then, based on the music sequence, a language model provides plausible utterance between a system and a user.

While the recent methods and their datasets have initiated developing and evaluating conversational music recommendation systems, there is room for improvements on various aspects. First, the two-stage process is an convenient design choice to generate the data and does not resemble a realistic scenario of conversations between a music recommender and a user. For example, at turn 1, the language model already completely knows the future music sequence, which could affect the utterances of both the system and the user. Second, none of the methods are multimodal, unlike how listeners

---

[1]TalkPlayData 2 and its generation code will be open-sourced after the review stage.

perceive and consume music in the real world. Third, there is not any component for personalization. Fourth, those datasets lack of extra labels or information such as user preferences on the recommendation or any reasoning step behind the recommendations, both of which can be crucial components in modern recommendation systems. Fifth and finally, in each dataset, all the conversations are generated based on the same assumptions – determining the music sequence is done by the same logic, and generating the utterance is done with the same prompt. This could lead to a mode collapse, i.e., every conversation may represent the same music recommendation scenario.

Those limitations motivate the development of the proposed pipeline and the datasets: TalkPlayData 2. The goal of TalkPlayData 2 is to provide conversation data for music recommendation research that covers *various conversation scenarios* and involves *multimodal* aspects of music. The architectural design choice is made to generate the data in a realistic scenario that resembles the real-world recommendation. TalkPlayData 2 also includes the user preference and reasoning messages for each turn, enabling to optimize a system not only to mimic the data, but also to maximize the user satisfaction. Finally, TalkPlayData 2 provides basic user profiles for each conversation.

## 2 CORE IDEA

$$y = f_\theta(x_{profile}, x_{goal}, x_{music}) \qquad (1)$$

A simplified formulation of the creation process for a data point of TalkPlayData 2 is Equation 1, $f$ is the creation pipeline, $\theta$ indicates the model weights of the involved LLM, $x_{profile}$ is a listener's demographic information and preferences, $x_{goal}$ represents the conversation objectives and scenarios, and $x_{music}$ is a set of music data, covering text, audio, and image modalities. $y$ indicates the outcome, a multi-turn conversation between the listener and the recommendation system about music discovery, consisting of queries, music items, and responses.

### 2.1 GROUNDING MUSIC DATA, $x_{music}$

To achieve factual data generation, TalkPlayData 2 is primarily based on a source dataset for the details about music items, $x_{music}$. In other words, $y_{music}$, the recommended tracks, must be merely a result of sequential selections of $x_{music}$, the recommendation pool. More details about the source data are provided in subsection 3.1. The music data $x_{music}$ is a set of loosely relevant music items, e.g., music tracks in a listening session in this paper (equivalent to playlists as in Doh et al. (2025)). It is provided with a rich set of metadata, tags, lyrics, audio, and images.

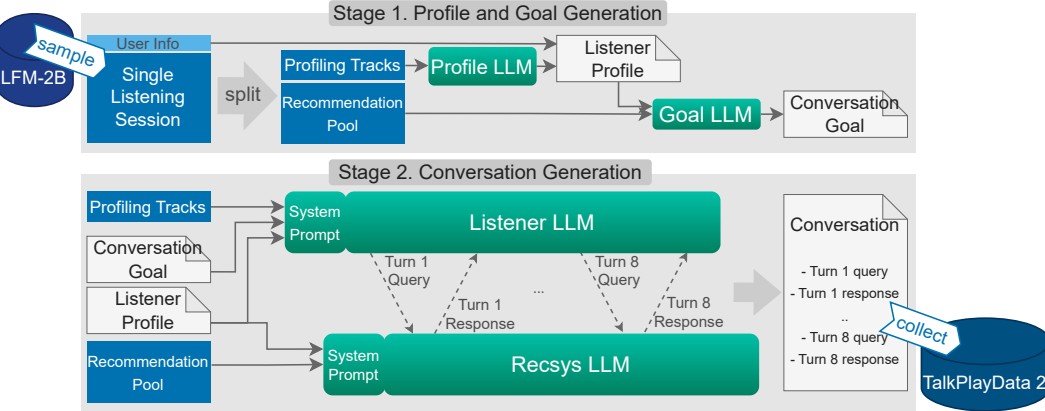

Figure 1: Overview of TalkPlayData 2 pipeline, consisting of four LLMs with specialized roles.

Table 1: LLM Capabilities Utilized in TalkPlayData 2 Generation

| Capability | Details |
| --- | --- |
| *Musical Aspect* | |
| Entity Recognition | Names of artists, albums, and tracks (Hachmeier & Jäschke (2024)) |
| Domain Knowledge | Key, chord, and tempo (Zhou et al. (2024); Li et al. (2024b)) |
| User Simulation | User profiles and goals (Zhang et al. (2025)) |
| *Multimodal Understanding* | |
| Text | Lyrics, tags (Vasilakis et al. (2024)), conversations (Kwon et al. (2024)) |
| Audio | Music audio signals (Gardner et al. (2023)) |
| Image | Album art images (Hayashi et al. (2024)) |
| *Agentic Interaction* | |
| Instruction Following | Adhering to recommendation constraints and producing responses |
| Goal Achievement | Achieving provided goals through multi-turn conversations |
| Chain-of-Thought | Generating intermediate 'thought' sentences to reflect and plan |

## 2.2 Data Generation Agents, $f = \{g_{goal}, g_{profile}, g_{listener}, g_{recsys}\}$

The creation system $f$ is formed by multiple LLM sessions, i.e., a set of agents, each of which has a role – as a listener profiler (Profile LLM, $g_{profile}$), a conversation goal setter (Goal LLM, $g_{goal}$), a listener (Listener LLM, $g_{listener}$), and a recommendation system (Recsys LLM, $g_{recsys}$). This agentic approach presents several advantages compared to relying on a single LLM session such as Doh et al. (2025). Most critically, it systematically prevents the agent from 'cheating' by looking at the data provided to other roles. This makes the data generation process highly realistic. For example, a conversation goal is shared with the Listener LLM so that it generates relevant queries to the Recsys LLM and achieves its goal. However, the goal is not shared with the Recsys LLM, whose role is to guess the goal of the Listener LLM through conversation. This approach also provides the role, task, and behavior instructions to each LLM with high clarity, leading to generate high-quality data. This is well-aligned with the recent multi-agent systems such as Fellowship of the LLMs (Arif et al., 2024), MAG-V (Sengupta et al., 2024), and AgentSGEN (Xuan et al., 2025), where dedicated roles improved semaantic fidelity and reduces mode collapse.

## 2.3 The Utilized Capabilities of LLMs, $\theta$

Besides grounding music data, the generation process of TalkPlayData 2 fully relies on various capabilities of LLMs encoded in its weight $\theta$, as in Table 1. Not only do the text-based capabilities need to be strong, but the multimodality of LLMs is also essential for the successful creation of TalkPlayData 2 to simulate recommendation systems and listeners who can see and listen to music. Table 1 summarizes the details of important capabilities, which are unblocked by the recent progress of LLMs.

## 2.4 User Profile and Conversation Goal, $x_{profile}$ and $x_{goal}$

Although the LLM generation process is often stochastic, it is well-known that naively sampling multiple times does not lead to diversifying the generation outcomes. Rather, a mode collapse often occurs, where the generated texts become too similar to each other in style and logic (Wang et al. (2025); Chen et al. (2025)). We observed this in our preliminary experiment - even when using different music items, the generated conversations had very similar styles. To address this issue, we utilize user demographic information and pre-defined conversation goals to generate diverse conversations. Furthermore, utilizing Goal LLM and Profile LLM, we enhance $x_{profile}$ and $x_{goal}$ to be more suitable for recommending music $x_{music}$, enabling the generation of more diverse and realistic conversations. The details are provided in subsection 3.2.

---

**Algorithm 1** Data Generation Process for Multi-modal Music Recommendation Conversations

---

**Require:** Listening sessions $S$, Set of tracks $M$, User Profile $U$, Conversation Goal $G$, Query message $Q$, Recommended Music $M_t$, Response $R$, Thought $T$, Progress Towards Goal $P$

1: **for** each listening session $s$ with $|M| \geq 21$ tracks **do**
2:      $M_{profile} \leftarrow$ sample 5 tracks of $M$
3:      $M_{pool} \leftarrow$ sample 16-32 tracks of $M - M_{profile}$
4:      $U_{base} \leftarrow$ user demographic information of $s$ (age, country, gender)
5:      $G_{base} \leftarrow$ sample 3 templates from goal dictionary
6:      $U_{final} \leftarrow$ ListenerProfileLLM($U_{profile}, U_{base}$)
7:      $G_{final} \leftarrow$ ConversationGoalLLM($G_{base}, M_{pool}, U_{final}$)
8:      $Q_1 \leftarrow$ ListenerLLM($U_{final}, G_{final}, M_{profile}$)      ▷ First query message ($Q_1$)
9:      $T_1^t, M_1, R_1 \leftarrow$ RecsysLLM($P_{final}, T_{pool}, Q_1$)      ▷ First track ($M_1$), response ($R_1$)
10:      $conversation \leftarrow [Q_1, M_1, R_1]$
11:      **for** turn $t = 2$ to $8$ **do**
12:          $P_t, T_t^l, Q_t \leftarrow$ ListenerLLM($U_{final}, G_{final}, M_{profile}, conversation$)
13:          $conversation$.append($Q_t$)
14:          $T_t^r, M_t, R_t \leftarrow$ RecsysLLM($U_{final}, M_{pool}, conversation$)
15:          $conversation$.append($M_t \ R_t$)

---

## 3 THE CREATION STEPS OF TALKPLAYDATA 2

### 3.1 OVERVIEW

**Base Dataset** The foundation of TalkPlayData 2 is the LFM-2b dataset (Schedl et al. (2022)), which provides session-based music listening history data of over 120,000 users spanning more than 15 years (February 2005 to March 2020). Beyond basic metadata (track, album, artist name), LFM-2b provides rich additional information including user demographic data (country, gender, age), last.fm genre/style annotations, and ID mappings to Spotify track identifiers. Additional multimodal information is acquired through the provided Spotify track identifiers and the API – preview audio snippets, album art images, release dates, and popularity metrics (as of 2025 July). Finally, we used pretrained music information retrieval models to estimate rich information: Madmom (Böck et al. (2016)) for tempo, key, and chords as well as Whisper (Radford et al. (2023)) for lyrics.

**Data Split** To reflect real-world recommendation scenarios, we performed a chronological data split. Sessions after 2019 were reserved for testing, while earlier sessions were used for training. To create the multimodal conversation dataset, we filter listening history sessions to include only tracks with Spotify track identifier mappings. Furthermore, to address cold-start user and cold-start item scenarios, we carefully sampled the test set conversations. Out of 1,000 test set conversations, 800 were sampled from the warm user pool, while 200 were sampled from the cold user pool. During sampling, we ensured balanced sampling across demographic attributes - country, gender, and age groups. This balanced sampling helps evaluate the model's performance across diverse user segments. Detailed statistics are provided in section 4.

**Large Language Models** The Google Gemini 2.5 Flash (`gemini-2.5-flash`, Comanici et al. (2025)) is chosen to create TalkPlayData 2. Google Gemini is the only available LLM API that supports the three modalities of TalkPlayData 2, and their models have demonstrated strong music understanding capabilities in various benchmarks (Ghosh et al., 2025; Carone et al., 2025a;b; Lee et al., 2025; He et al., 2025; Kumar et al., 2025; Ma et al., 2025). In the preliminary experiments, there seem noticeable performance gaps between Gemini 2.5 Flash and its 'Lite' version when it comes to music understanding. The 2.5 Pro version, the most advanced version of Google Gemini as of 2025 Aug, was not chosen for two reasons: it is about 3-4 times more expensive, and a more advanced LLM is needed for the LLM-as-a-judge evaluation (subsection 4.3).

### 3.2 GENERATION PROCESS

The overall process iterates over the listening sessions $S$ and converts each session $s \in S$ into a conversation. Each session $s$ consists of a list of tracks $M$ and basic user demographic information

$U$, including age group, gender, and country. For each conversation, we require sessions containing at least 21 tracks to ensure a sufficient recommendation pool. From each session, 5 tracks are sampled as profiling tracks ($M_{profile}$) to inform the conversation style sampling, while another 16-32 tracks form the recommendation pool ($M_{pool}$).

For a single conversation, four separate LLM sessions are created, each of which corresponds to $g_{profile}$, $g_{goal}$, $g_{listener}$, and $g_{recsys}$, respectively. Their instructions are carefully designed after many iterations of reviews and updates to ensure correct behaviors and response formats. As outlined in Algorithm 1 and Figure 1, the overall generation process consists of two stages: i) profiling and goal generation, and ii) conversation generation. During the first stage (lines 1-8), we create a user profile $U_{final}$ based on profiling tracks $M_{profile}$ and demographic information $U_{base}$; and a conversation goal $G_{final}$ from base goal templates $G_{base}$, recommendation pool $M_{pool}$, and user profile $U_{final}$. This customization is responsible for diversifying the conversation in style. The conversation generation stage (lines 11-15) follows with alternating API calls between the listener and the recommendation systems, where queries $Q_t$, music recommendations $M_t$, and responses $R_t$ build upon the conversation history while maintaining coherence with the conversation goals $G_{final}$. The detailed instructions and responses for the LLMs are provided in Appendix B.

**Listener Profile LLM**    The role of the Listener Profile LLM is to analyze the profiling tracks and infer high-level preference information of the listener, given demographic information, such as gender, age group, and country. During generating TalkPlayData 2 as well as using it, this information provides the personalization aspect, which is crucial in modern recommendation systems and music consumption (Schedl et al. (2021); Kaminskas & Ricci (2012); North & Hargreaves (2008)). The LLM combines the provided demographic profile with the track analysis to estimate musical preferences including preferred musical culture, top artist, and top genre. All the text data (metadata, tags, and lyrics), audio, and image data described in subsection 3.1 are provided to the LLM.

**Conversation Goal LLM**

The Conversation Goal defines the session-level goal that the listener wants to achieve through conversation with the recommendation system. To guarantee the overall diversity of the conversations in TalkPlayData 2, a diverse set of conversation goals is needed; while each conversation goal should be plausible given the recommendation pool (Li et al. (2024a)). Before the generation process, a set of 44 conversation goal templates is prepared. A template is defined by two properties, the topic and the specificities. In total, 11 topics are defined to cover various types of multi-modal music discovery conversations, as listed in Table 3. These topics decide which aspect of music the conversation will be based on and cover recommendation scenarios based on audio (Deldjoo et al. (2024); Van den Oord et al. (2013), lyrics (Patra et al. (2017); Vystrčilová & Peška (2020)), visual information (Saito & Itoh (2011); Libeks & Turnbull (2011)), and emotion (Han et al. (2010)). The specificities define how specific the query and the target music are, resulting in 4 cases as in Table 3. This two-dimensional formalization of LL, HL, LH, and HH provides a structured view of recommendation scenarios such as exploratory search (Marchionini (2006); Schedl et al. (2015)), lookup tasks (Marchionini (2006)), and query granularity (Sun & Zhang (2018); Jannach et al. (2021)).

Table 2: Conversation Goal Axis 1 - Topics

| Code | Description | Example |
|------|-------------|---------|
| A | Audio-Based Discovery | "Discover songs with immersive soundscapes" |
| B | Lyrical Discovery | "Songs about love" |
| C | Visual-Musical Connections | "Music that looks colorful and vibrant" |
| D | Contextual & Situational | "Music for working and studying" |
| E | Interactive Refinement | "Let's play hard rock and transit to modern rock" |
| F | Metadata-Rich Exploration | "Find multiple songs from the Hamilton musical" |
| G | Mood & Emotion-Based | "I need something to cheer me up" |
| H | Artist & Discography Discovery | "Tell me about this artist's other works" |
| I | Cultural & Geographic | "Music from Alaska" |
| J | Social & Popularity Context | "What's trending right now?" |
| K | Temporal & Era Discovery | "Music from the 80s please" |

Table 3: Conversation Goal Axis 2 - Specificities

| Code | Description | Example |
|------|-------------|---------|
| LL | Low query specificity
Low target specificity | "Play some chill music"
(Many tracks are possible as a successful recommendation) |
| HL | High query specificity
low target specificity | "Find bebop jazz with saxophone, 1950s-60s"
(Many tracks are possible, the query is somewhat specific) |
| LH | Low query specificity
high target specificity | "What was the popular song from a recent musical movie?"
(One or few tracks are possible, the query is not specific) |
| HH | High query specificity
high target specificity | "Windup by Hayoung Lyou, the jazz composer and pianist"
(One track is possible, the query is highly specific) |

There are two steps to generate a conversation goal. First, three templates are randomly sampled. They decide the potential directions, but they are still template candidates, since some of them may not be plausible per the recommendation pool. For example, the recommendation pool may consist of all instrumental music, which would limit lyric-based conversations. Second, the three base conversation goals are fed to the Goal LLM ($g_{goal}$), whose role is to select the most plausible goal based on the recommendation pool and customize the overall goal with concrete examples.

To improve conversation pacing and realism, each conversation goal includes a target turn count that guides the expected resolution time: HH specificity goals target 1-2 turns (quick resolution), HL specificity targets 3-4 turns (moderate exploration), LL specificity targets 3-7 turns (extensive exploration), and LH specificity targets 6-8 turns (detailed exploration). The target turn count is determined by the Goal LLM ($g_{goal}$) based on goal complexity and recommendation pool content. While conversations always continue to the full 8 turns for consistent training data, the target turn count influences the listener's pacing strategy and goal achievement approach.

**Listener LLM and Recsys LLMs**   Based on the profiling tracks, the conversation goal, and the listener profile, the conversation is initiated by the Listener LLM ($g_{listener}$). On Recsys LLM ($g_{recsys}$), after being initialized with the listener profile and the recommendation pool (but not the conversation goal), it starts to respond to the Listener LLM's initial query. In the subsequent turns (from Turn 2), the Listener LLM actively engages with the recommended music by listening to the audio samples and seeing the album artwork before formulating responses. This multimodal interaction allows the Listener LLM to provide more nuanced and informed feedback about the recommendations, to which Recsys LLM then makes subsequent recommendations. The Listener LLM also labels whether the Recsys LLM's recommendation is making positive progress towards achieving the goal. This information is expected to be used as a 'preference' signal during training recommendation models using reinforcement learning, or as an auxiliary classification target. Finally, in every turn, both the Listener and the Recsys LLMs generate 'thought' before generating their response message to each other. A 'thought' is used to analyze the input message (from the other LLM), increasing interpretability during both dataset creation and utilization.

## 4   TALKPLAYDATA 2: STATISTICS AND EVALUATION

### 4.1   STATISTICS

Table 4 summarizes the key statistics of TalkPlayData 2, which consists of a 15,199 training set and 1,000 test set divided using chronological splitting to reflect real-world deployment scenarios. The test set includes 129 cold users and 2,982 cold tracks for evaluating cold-start scenarios, while a distinctive characteristic is the comprehensive inclusion of user queries, assistant responses, and detailed thought processes that provide valuable insights into the reasoning behind preferences and recommendations, enabling interpretable music recommendation systems.

Table 4: TalkPlayData 2 statistics

| Counts of | Training | Evaluation |
|-----------|----------|------------|
| Conversations | 15199 | 1000 |
| Warm Users | - | 371 |
| Cold Users | - | 129 |
| Total Users | 8591 | 500 |
| Warm Tracks | - | 3779 |
| Cold Tracks | - | 2982 |
| Total Tracks | 43597 | 6761 |

Table 5: Comparison among conversational music recommendation datasets. Gray text indicates closed-source datasets. CS refers to cold-start Split.

| Dataset | Profile | Goal | Thought | CS | Conv. | Track | User | Turns |
|---|---|---|---|---|---|---|---|---|
| JAMSessions | ✓ | ✗ | ✗ | N/A | 112K | 100k | 104K | 1.00 |
| Text2Track | ✗ | ✗ | ✗ | N/A | 1M | 500K | - | 1.00 |
| CPCD | ✗ | ✗ | ✗ | ✗ | 0.1K | 107K | - | 5.70 |
| LP-MusicDialog | ✗ | ✗ | ✗ | ✗ | 288K | 391K | - | 4.97 |
| TalkPlayData 1 | ✗ | ✗ | ✗ | ✓ | 532K | 406K | - | 6.95 |
| TalkPlayData 2 (**Ours**) | ✓ | ✓ | ✓ | ✓ | 16.2K | 47K | 9k | 8.00 |

Table 5 presents a comparison among conversational music recommendation datasets: including two closed-source single-turn conversation datasets (Palumbo et al., 2024; Melchiorre et al., 2025), a human conversation dataset (Chaganty et al., 2023), and two LLM-based synthetic datasets (Doh et al., 2024; 2025). While TalkPlayData 2 does not have a large number of conversations, it represents the dataset most similar to real music recommendation scenarios, featuring 1) long conversation turns, 2) user profiles, 3) conversation goals, 4) chain-of-thought, and 5) cold-start splits.

## 4.2 HUMAN EVALUATION

We assess the quality of our generated data through human evaluation, focusing on two key aspects: 1) *relevance* – determining the alignment between the retrieved music items and the user query, and 2) *naturalness* – assessing the likelihood of such a conversation occurring in real life. We adhere to a mean opinion score that uses a 5-point Likert scale. A total of 26 raters evaluated 10 randomly sampled dialogues each, resulting in 260 total ratings. For comparison models, we select open-source conversational music recommendation datasets. CPCD (Chaganty et al. (2023)) is a human conversation dataset about music, and LP-MusicDialog (Doh et al. (2024)) and TalkPlayData 1 (Doh et al. (2025)) are synthetic conversation datasets generated by a single LLM.

As shown in Table 6, TalkPlayData 2 achieves the highest scores in both dimensions. The high relevance score demonstrates the effectiveness of our multimodal approach, where LLMs consider both audio and visual aspects of music during recommendation, leading to more accurate and contextually appropriate suggestions compared to text-only approaches (Doh et al. (2024; 2025)). The strong naturalness score highlights the effectiveness of our multi-LLM framework. By orchestrating interaction between the Conversation Goal LLM and the Profile LLM, the system enables naturalistic exchanges between the Listener LLM and the RecSys LLM, thereby improving both conversational coherence and user simulation. These results suggest that our approach of using multiple LLMs creates more engaging and effective conversational recommendations compared to both human conversations (Chaganty et al. (2023)) and single-LLM approaches (Doh et al. (2024; 2025)).

Table 6: Comparison of conversational music recommendation datasets. Type stands for the subject of conversation. Relevance and Naturalness show Mean Opinion Scores of 5 Likert Scale.

| Datasets | Type | LLMs | Multimodal | Relevance | Naturalness |
|---|---|---|---|---|---|
| CPCD | Human | - | - | 4.08 | 4.01 |
| LP-MusicDialog | Synthetic | 1 x ChatGPT | ✗ | 3.90 | 3.95 |
| TalkPlayData 1 | Synthetic | 1 x Gemini-1.5-Flash | ✗ | 4.04 | 4.01 |
| TalkPlayData 2 | Synthetic | 4 x Gemini-2.5-Flash | ✓ | 4.11 | 4.15 |

## 4.3 LLM-AS-A-JUDGE EVALUATION

An LLM-as-a-judge evaluation is conducted on its test set to provide a detailed analysis of the design choices of TalkPlayData 2 (Zheng et al. (2023); Chen et al. (2024)). It plays a crucial role in two aspects: 1) quality control during dataset generation and 2) a cost-effective alternative to human evaluation. While human evaluation is considered the gold standard (reported at Section 4.2), it is often impractical for large conversational datasets due to cost and scale limitations.

Table 7: Evaluation Results Summary

| Evaluated Entity | Focused Aspect | Aggregated Score | Score Distribution (1-4) |
|---|---|---|---|
| Conversation Goal | Plausibility given recommendation pool | 3.93/4 | |
| Listener Profile | Appropriateness | 3.41/4 | |
| Chat Element (Listener) | progress_towards_goal: Label accuracy | 3.38/4 | |
| | thought: Overall quality | 3.98/4 | |
| | message: Linguistic quality | 4.00/4 | |
| | message: Helpfulness towards goal | 4.00/4 | |
| Chat Element (RecSys) | thought: Overall quality | 3.52/4 | |
| | track_id: Recommendation quality | 3.35/4 | |
| | message: Linguistic quality | 3.69/4 | |
| | message: Alignment with track | 3.83/4 | |

For the judge LLM, Gemini 2.5 Pro is used, which is a more advanced model than the one used in the generation process. Although the self-referential bias may affect (Wataoka et al., 2024), it was chosen because the Gemini family is the only available models that supports three modalities through APIs. For each conversation, multiple calls are made to the judge LLM, each of which is asked to evaluate a specific aspect of the conversation, with an appropriate instruction, scoring criteria, and response format. When the track information is needed, the judge LLM is provided with all the textual, audio, and image data of the tracks as done in the generation process. To further address this issue, we provide a separate LLM-as-a-judge result that resembles the human evaluation in subsection 4.2 at the end of this section.

Table 7 summarizes the evaluation results across different aspects of conversation quality. Among the focused aspects, some of them are simply better to be higher since they would provide information used during training, e.g., progress_towards_goal or thought. Some others are not always the case: e.g., although we pursue high linguistic quality in TalkPlayData 2, it may be part of the scope of training a conversational recommendation system that can handle queries with incorrect grammar and unclear instructions. This is also discussed in the following analysis.

**Conversation Goal** This is evaluated on the plausibility of the goal given the recommendation pool, focusing on the behavior of Goal LLM. The high average score of 3.93/4 indicates the effectiveness of the proposed setup of sample, select, and customize. Additionally, the distributions are reasonably balanced over the specificity (22%, 34%, 28%, 16%) and the category (9%, 18%, 11%, 11%, 12%, 9%, 11%, 16%, 3%), respectively, along the codes in Table 2 and Table 3.

**Listener Profile** This is evaluated on the appropriateness of the user profile given the profiling tracks. Its high average score of 3.41/4 indicates that the profile generated by the Profile LLM is mostly well-aligned with the profiling tracks.

**Chat Element** On the Listener LLM, the progress_towards_goal is evaluated on its accuracy; if the Listener LLM's binary label on whether the recommended track moves the conversation towards the goal is correct. A high accuracy is desirable, ensuring the credibility of progress_towards_goal, which can be used as user feedback when training an LLM recommendation system. The score of 3.38, with over 75% of the conversations being evaluated as a score of 4.0, 'Excellent', indicates that it is well-labeled. The thought is evaluated on overall quality including coherence, alignment, helpfulness, and consistency. The high average score of 3.98/4 indicates that the thoughts are well-written and can be used during training for explanationability, or as a chain-of-thought. The message is evaluated on two orthogonal aspects. First, in its linguistic quality including naturalness, realism, and consistency, the average LLM judge score is very high – 4.00/4. Second, in its utility (helpfulness towards goal), the average score is also 4.00/4. Overall, the Listener LLM's chat elements are well-written, and helpful.

On the Recsys LLM, the thought is evaluated on overall quality including coherence, alignment, helpfulness, and consistency. The high average score of 3.52/4 indicates that the thoughts are well-written and can be used during training for explanationability, or as a chain-of-thought. The track_id

is evaluated on its recommendation quality – the relevance between the user query and the recommended track. The score of 3.35/4 indicates in TalkPlayData 2, the Recsys LLM selects highly relevant items to each query most of the time (score of 4 for 73% ). The message is evaluated on two orthogonal aspects. First, in its linguistic quality including naturalness, realism, and consistency, the average LLM judge score is 3.69/4, which is slightly lower than the Listener LLM's score but still high. Second, in the accuracy of the track information with respect to the recommended track, the average score is 3.83/4, validating that the Recsys LLM provides accurate track information in its message most of the time.

## 4.4 ABLATION STUDY AND COMPARISON WITH EXISTING DATASETS

Table 8: KL divergence to uniform ($KLD_u$, ↓) and coverage (↑) across ablations.

|  | $KLD_u$ (Specificity) | $KLD_u$ (Topic) | Coverage (Specificity) | Coverage (Topic) |
|---|---|---|---|---|
| TalkPlayData 2 | 0.240 | 0.110 | 1.000 | 1.000 |
| A1: no goal | 0.316 | 0.700 | 0.750 | 0.455 |
| A2: no profile | 0.553 | 0.045 | 0.750 | 1.000 |
| A3: no goal+profile | 0.395 | 0.451 | 0.750 | 0.455 |
| CPCD | 1.015 | 0.727 | 0.750 | 0.636 |
| LP-MusicDialog | 1.003 | 1.560 | 0.500 | 0.364 |
| TalkPlayData 1 | 1.386 | 1.639 | 0.250 | 0.364 |

The goal of the ablation study is to provide empirical evidence that analyzes and supports the design choices of TalkPlayData 2. The experiments are conducted with three configurations: removing the conversation goal (A1), the listener Profile (A2), and both (A3). In each configuration, 50 conversations are generated in total, all based on the same base data subset of LFM-2b. Then, an LLM judge is prompted to classify each conversation into one of the 4 specificities and 11 topics. We use i) the KullbackLeibler divergence (KLD) to measure how close the empirical distributions are to a uniform distribution and ii) Coverage, the proportion of classes that have non-zero items, to track any potential strong category biases.

As presented in Table 8 (top rows), only the proposed pipeline achieved the full coverage on Specificity and Topic, as well as the lowest $KLD_u$ in Specificity and the second lowest $KLD_u$ in Topic. In detail, first, the overall importance of the goal is clear, based on the significant degradation from every aspect in A1 and A3. This is expected, since the goal is the only prompt where the Specificity and the Topic are defined. Second, the impact of the profile is more nuanced, since in A2, the metrics on the Topic do not indicate any issues, while the diversity of the Specificity shows severe regression, i.e. the goal alone is enough to generate conversations with diverse topics, but not with diverse specificities. We conjecture this is due to the close relationship between the Specificity and listener behaviors, as well as the sequential order that the profile conditions the goal. The profile includes not only demographics but also open-vocabulary attributes such as preferred musical culture, top artists, and genres – altogether, seemingly contributing to the diversity of how specific a user would query and expect; and when there is a lack of such information, the Goal LLM is biased towards certain specificities.

The KLD and the Coverage are also measured on the existing datasets, as in the bottom rows of Table 8. In both metrics and the axes, the existing datasets exhibit a low diversity. As mentioned in section 1, this result reminds the motivation for TalkPlayData 2 – that without diverse prompts, LLM-based data generation often suffers from a severe mode collapse, shown by the particularly high KLD values of TalkPlayData 1 and LP-MusicDialog.

## 5 CONCLUSION

In this paper, we introduced TalkPlayData 2, a new multimodal dataset for conversational recommendation systems. In the data generation pipeline, separate LLM calls are first made to create a listener profile and a conversation goal for each conversation. Using them as a condition, two separate LLMs talk to each other under the role of a music listener and a music recommendation system. The conversation is conducted for 8 turns, and the data is collected as a conversation. Notably,

all the LLMs are multimodal, enabling to generate conversation with multimodal aspects of music being considered. The LLMs have access to different subsets of the information, a design choice that is highly similar to the real-world conversational recommendation systems. In the evaluation, we conducted an LLM-as-a-judge evaluation as well as a human evaluation, which shows that Talk-PlayData 2 is a promising dataset for training and evaluating conversational music recommendation systems.

There are still many interesting directions to explore in the future. First, the in-context recommendation of the Recsys LLM has a limitation in the number of tracks it can consider. Expanding its recommendation pool size is a natural direction. Second, although TalkPlayData 2 consists of highly natural conversations, it is still limited in various aspects including the speaking style and language. Third, due to the cost of the LLMs, during the data generation, only a short audio snippet and a small album cover image are used. Using longer audio and more diverse visual information (such as music videos, live performances, and any other modalities) can make the data even more deeply multimodal, enabling holistic multimodal conversational recommendation systems.

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

## A  ADDITIONAL ANALYSIS OF TALKPLAY2 DATASET

**Generation Cost Analysis**  The generation process utilizes four distinct LLM components with varying computational requirements. For 1,000 conversation data, the RecSys LLM consumes the highest token count at 171.9M tokens (66.8% of total), followed by the Listener LLM at 64.7M tokens (25.1%), Goal LLM at 17.5M tokens (6.8%), and Profile LLM at 3.2M tokens (1.2%). The multimodal processing follows fixed token allocations: each image consumes 258 tokens (300Œ300 pixels), and each audio segment consumes 96 tokens (3 seconds at 32 tokens/second). The total generation cost amounts to $109.08 for 1,000 conversations.

## B  API SPECIFICATIONS OF THE LLMS DURING GENERATION

During the generation process, the LLM calls consist of many long prompts, defining the task, behavior, input data, and the response format. In this Appendix, we provide a summary of the prompt as follows.

### B.1  LISTENER PROFILE LLM

**Input (demographic information and list of text and track entities)**

```
[f"You are an expert in music and demographic analysis. Given the demographic profile below and
tracks, please analyze the tracks and infer the most representative preferred_musical_culture,
artist and genre that define this listener's taste.",
Demographic Profile:
- age_group: [factual age group]
- country: [factual country]
- gender: [factual gender]
- preferred_language: [factual language]
"Title: [track 1 title], Artist: [track 1 artist], ...", AudioContent, ImageContent, ...
..., "Title: [track 5 title], Artist: [track 5 artist], ...", AudioContent, ImageContent]
```

**Output (YAML block of Listener Profile)**

```
preferred_musical_culture: [most representative musical culture from tracks]
top_1_artist: [most representative artist from tracks]
top_1_genre: [most representative genre from tracks]
```

### B.2  CONVERSATION GOAL LLM

**Input (list of text and track entities)**

```
[f"You are an expert in music listening ... Step 1: Analyze the tracks and the provided
conversation goals templates, and select the most appropriate conversation goal ... Step 2:
Generate a new conversation goal that is more specific to the tracks, based on the selected
conversation goal.",
"Title: [track 1 title], Artist: [track 1 artist], ...", AudioContent, ImageContent, ...
..., "Title: [track 32 title], Artist: [track 32 artist], ...", AudioContent, ImageContent,
f"Here are the conversation goal templates, based on which you will generate the new conversation
goal: {{three_conversation_goal_templates}}"]
```

**Output (YAML block of Conversation Goal, some are omitted for brevity)**

```
category_code: [alphabetical topic code among A-K]
specificity_code: [one of LL, HL, LH, HH]
target_turn_count: [1-8 based on specificity code]
listener_goal: [customized goal description for the tracks]
listener_expertise: [description of the listener expertise]
initial_query_example_1: [plausible initial query example 1]
```

## B.3 LISTENER LLM (FIRST TURN)

**Input (list of text and track entities)**

```
[f"You are an AI assistant role-playing as a music listener. Your personality, knowledge, and
objectives are STRICTLY defined by the Listener Profile and Conversation Goal provided below.
... For your very first message (Turn 1), ... choose one of the initial query examples provided
in the Conversation Goal and use it ...",
"Title: [profile track 1 title], ...", AudioContent, ImageContent, ...
..., "Title: [profile track 5 title], ...", AudioContent, ImageContent,
f"{{listener_profile}}", f"{{conversation_goal}}",
f"You are starting a new music discovery conversation. ... Turn 1. Now, create ... first turn
query to RecSys ..."]
```

**Output (YAML block of Listener's First Message)**

```
thought: [internal reasoning about the goal and approach]
message: [natural opening message to the recommendation system]
```

## B.4 RECSYS LLM (FIRST TURN)

**Input (list of text and track entities)**

```
[f"... You are TalkPlay, an expert music recommendation system with deep musical knowledge,
audio analysis capabilities, and image analysis capabilities. ... You MUST recommend
ONLY from the provided available tracks. ... Make personalized music recommendations ...
{{listener_profile}}",
"Title: [pool track 1 title], ... ID: [pool track 1 id], ...", AudioContent, ImageContent, ...
..., "Title: [pool track 32 title], ... ID: [pool track 32 id], ...", AudioContent, ImageContent,
"... Turn 1. ... Listener's message: {{listener_message}} ... "]
```

**Output (YAML block of Recsys Response)**

```
thought: [analysis of listener's request and selection reasoning]
track_id: [selected track identifier from the pool]
message: [natural response with track information and explanation]
```

## B.5 LISTENER LLM (SUBSEQUENT TURNS)

**Input (list of text and track entities)**

```
[f"Title: [recommended title], Artist: [recommended artist], ... ", AudioContent, ImageContent,
f"You just listened to this recommended track: ... The recommendation system said:
'{{recsys_message}}' ... Assess whether this track moves you toward achieving your Conversation
Goal ... "]
```

**Output (YAML block of Listener's Response)**

```
thought: [internal evaluation of the track and strategy]
goal_progress_assessment: [MOVES_TOWARD_GOAL or DOES_NOT_MOVE_TOWARD_GOAL]
message: [feedback and next request toward the goal]
```

### B.6  RECSYS LLM (SUBSEQUENT TURNS)

**Input (list of text and track entities)**

```
[f"... Previous Tracks: {{used_track_ids}} ... Listener's message: '{{listener_message}}' ...",
"... NO DUPLICATES ... Maintain conversation coherence and respond naturally "]
```

**Output (YAML block of Recsys Response)**

```
thought: [analysis of feedback and next recommendation strategy]
track_id: [next selected track identifier]
message: [response with new track and reasoning]
```

## C  VALIDATING THE LLM-AS-A-JUDGE

### C.1  CROSS-DATASET RANKING CORRELATION STUDY

A cross-dataset validation study is conducted to address concerns regarding the reliability of LLM-as-a-judge and potential self-enhancement bias (Li et al., 2025; Wataoka et al., 2024). In this study, the judge LLM (Gemini 2.5 Pro) is prompted to evaluate 30 randomly sampled conversations; from TalkPlayData 2 and the three baseline datasets (CPCD, LP-MusicDialog, TalkPlayData 1). Two prompts are used to score Relevance and Naturalness, mirroring the human evaluation in subsection 4.2. The prompts also include requesting to respond with reasoning for its scoring.

Table 9: LLM-as-a-Judge scores on TalkPlayData 2 and baseline datasets on a 4-point Likert scale.

| Dataset | Relevance | Naturalness |
|---|---|---|
| CPCD (Human-authored) | 3.24 | 2.50 |
| LP-MusicDialog | 2.81 | 2.78 |
| TalkPlayData 1 | 3.56 | 3.73 |
| TalkPlayData 2 | 3.45 | 3.56 |

For Relevance, the LLM-judge's scores in Table 9 align with the human evaluation in Table 6. Both humans and the LLM-judge place TPD1, TPD2, and CPCD in a high-quality cluster, and both identify LP-MusicDialog as the outlier with the lowest relevance. This demonstrates the judge is a reliable proxy for human perception of recommendation quality.

For Naturalness, the judge's ranking differs from the human ranking, and in doing so, the result disproves the concern of self-enhancement and referential bias (Wataoka et al., 2024). The judge gives the lowest naturalness score (2.50) to the real human-authored dataset (CPCD). A manual inspection of the judge's reasoning confirms it is correct to do so by consistently penalizing unnatural human phrases (e.g., in chat '1e6035d..', it flagged the awkward response "so I know what you are looking for?" to a user's query).

The judge's high score for TalkPlayData 1 (3.73) over TalkPlayData 2 (3.56) can be explained by the difference in the generation mechanisms. The utterance generation task for TPD1 (connecting a pre-determined music sequence with conversations) may be a linguistically simpler task, because a single language model (the Listener) with full information about the music sequence does not necessarily challenge itself (the Recsys) with impossible queries. Our inspection confirms that the LLM judge penalized TalkPlayData 2 's naturalness score primarily when "the system fails to fulfill direct user queries."

In summary, this study validates our LLM-judge as a reliable evaluator that i) correlates with human relevance rankings and ii) is a stricter and more holistic critic of naturalness than human raters.

# D    BASELINES

We present the official baseline results from the Conversational Music Recommendation Challenge of the EACL 2026 NLP4MusA workshop.[2] The task is defined as a two-stage pipeline: i a Recsys model retrieves candidate tracks, and ii) an LLM generates a natural language response. The primary evaluation metric is Normalized Discounted Cumulative Gain (nDCG) at k={1, 10, 20}, averaged across all conversation turns.

This official baseline result is solely provided by the task organizers and presented with mode details in Table 10. We present Table 10 only for convenience of the readers and this result should be referred by Epure et al. (2025a;b).

Table 10: Official Baseline Results for the EACL 2026 Task on the TalkPlayData 2 Test Set  (Epure et al., 2025a;b)

| Model | nDCG@1 | nDCG@10 | nDCG@20 |
|---|---|---|---|
| Random | 0.0000 | 0.0001 | 0.0002 |
| Popularity | 0.0005 | 0.0018 | 0.0024 |
| BERT + Llama-1B | 0.0038 | 0.0142 | 0.0189 |
| BM25 + Llama-1B | 0.0139 | 0.1015 | 0.1181 |

---

[2]https://sites.google.com/view/nlp4musa-2026

