# OpenReview forum: "TalkPlayData 2: An Agentic Synthetic Data Pipeline for Multimodal  Conversational Music Recommendation"
_ICLR.cc/2026/Conference — Submitted to ICLR 2026_

### Official Review · Reviewer_Zw5K · 2025-10-26

**Soundness:** 3
**Presentation:** 3
**Contribution:** 3
**Rating:** 6
**Confidence:** 3

**Summary:**

This paper presents TalkPlayData 2, a new synthetically constructed conversational recommendation dataset for music. The authors overview the design decisions behind their dataset and the agentic framework used to construct it, and evaluate its overall quality as a simulation for real music recommendation conversations.

**Strengths:**

- The overall documentation of dataset construction is exceedingly detailed. The authors attention to detail in describing all parts of the dataset generation pipeline are quite welcome, and strengthen the contribution exceedingly.
- The high level goal of the paper is quite timely, and seems to address a much needed gap in current datasets for training / evaluation of conversational music recommendation.

**Weaknesses:**

- From a writing perspective, while the finale 3-9 pages of the paper have quite good style, the first two pages stand out as remarkably different and somewhat overly sparse and incoherent relative to the rest of the text. The paper seems to be without a proper introduction, and thus, the beginning of this manuscript feels jarring and the motivation is left underdeveloped. While this may be helped by moving the related works to the forefront of the paper, it would be more useful to have a more standard introduction as typical of an ICLR paper.
- The main weakness of the paper, which aligns with the above issue, is that there is a lack of proper evaluation showing a real *use* for the dataset, as the present evaluation only assesses the quality of the synthetic data itself. While this is important, it would be useful to assess how MLLMs perform on the 0-shot test set of TalkPlayData 2, even if just a limited set of them, in order to establish a clear use for the dataset (in this case, evaluation of conversation music recommendation abilities).
- The objective evaluation, while concerted, feels as if it is missing a clear sort of baseline. It is both unclear whether the LLM-as-a-judge setting is even valid in the context (such a verification could be performed by a limited amount of human verification / user study to correlate LLM ratings with human ratings), and more importantly, how these numbers would compare against any sort of baseline data. While it is understandable that not all other datasets share the same construction as the present one, even comparing in the aspects that are shared with other datasets (such as TalkPlayData 1) would significantly improve the analysis here.

Overall, I believe while there are key issues in the paper, the depth of technical documentation outweighs these at least to cross the threshold for acceptance. However, I believe that by addressing the above concerns, the paper would be significantly improved.

**Questions:**

No real questions, please see weaknesses for main points.

---

> ### Author Response · Authors · 2025-11-17
>
> We thank Reviewer 4 for their constructive and insightful feedback. We agree with the suggested weaknesses. Based on their guidance, we have performed a major revision of the paper and run new experiments to address every point.
>
> ### 1. On W1 (Writing, Introduction, and Motivation):
>
> We thank the reviewer for this crucial feedback. You were correct that the original introduction was sparse and the motivation underdeveloped. We have performed a major revision of the paper's front matter based on this guidance.
>
> - We have written a new, comprehensive Section 1: Introduction that properly defines the task, explicitly lists the 5 key limitations of prior work (e.g., mode collapse, lack of multimodality, no personalization), and then frames TPD2's contributions as the direct solution to these problems.
>
> We believe the paper's structure, motivation, and clarity are now substantially stronger.
>
> ### 2. On W2 (Lack of "Real Use" / Benchmark):
> This is an excellent point regarding the paper's utility. We are pleased to report two major updates on this front, which have been added to the revised paper (Appendix).
>
> - **A)** TPD2 Selected for EACL 2026 Shared Task: TalkPlayData 2 has **been selected as the official dataset for the "Conversational Music Recommendation Challenge (Music-CRS),"** a shared task at the [EACL 2026 NLP4MusA workshop](https://sites.google.com/view/nlp4musa-2026/shared-task). This provides a strong validation of its "real use" and utility as a new, challenging benchmark for the community.
>
> - **B)** New Downstream Task Baselines: To directly address the request to "assess how MLLMs perform," we have **added the official baseline results** from this shared task to the Appendix. This new table establishes the first nDCG@k (k={1, 10, 20}) performance benchmarks on TPD2 for standard retrieval (BM25, BERT) and generative (Llama-3.2) models.
>
> ### 3. On W3 (Validating the LLM-Judge and Baselines):
> This was another critical insight. We have run two new validation experiments to address both parts of this concern. It is now included in the Appendix.
>
> - **A)** Validating the LLM-Judge (vs. Human Eval): To validate our judge and address concerns of bias (also raised by R1), we conducted a new **Cross-Dataset Ranking Correlation** study (now in the Appendix). We ran our judge (Gemini 2.5 Pro) on samples from TPD1, CPCD, and LP-MusicDialog.
>
>   - For **Relevance**, the judge's quality ranking strongly correlates with our human evaluators' ranking (from Table 6). Both humans and the LLM-judge identify LP-MusicDialog as the clear outlier.
>
>   - For **Naturalness**, the judge proved its objectivity and disproved any bias by scoring real human data (CPCD) lowest (2.50/4.0), correctly identifying awkward human phrasing.
>
>   - This experiment demonstrates our judge is a reliable and stringent proxy for human perception.
>
> - **B)** Comparative Evaluation (on Diversity): To provide a clear baseline for our pipeline, we ran our new diversity analysis (from Sec 4.4) on TPD1, CPCD, and LP-MusicDialog. The new results (in Table 8) are definitive: simpler, single-LLM approaches like TPD1 suffer from catastrophic mode collapse (e.g., KLD > 1.3, 25% specificity coverage), while our TPD2 pipeline achieves 100% coverage. This quantitatively proves our pipeline's novel design is necessary.
>
> - **C)** Comparative Evaluation (on chat element): As other datasets are structured significantly differently, it seems not easily doable to run LLM-as-a-judge evaluation (like Table 7) for baseline datasets. However, we sincerely hope the addition of Table 9 in the Appendix - cross-dataset ranking correlation study - and the discussion addressed the concern to some extent.
>
> We believe these significant additions (a new introduction, shared task nDCG baselines, and two new validation studies) have addressed your concerns and made the paper substantially stronger.

---

### Official Review · Reviewer_DgEG · 2025-10-30

**Soundness:** 2
**Presentation:** 2
**Contribution:** 1
**Rating:** 2
**Confidence:** 5

**Summary:**

The paper presents TalkPlayData 2, a synthetic multimodal conversational music recommendation dataset. It is generated with an agent pipeline of four LLM roles: a Profile model that builds a listener persona from demographic info and recent tracks, a Goal model that defines what the listener is trying to get from the conversation, a Listener model that simulates the user over multiple turns, and a Recsys model that recommends music. Each conversation is grounded in real listening sessions from LFM-2b. From each session, the pipeline samples a small pool of candidate tracks. All recommendations must come from this pool. For each track, the system includes metadata, lyrics, audio previews, and album art, so the models can refer to musical, lyrical, and visual qualities. The dataset is split chronologically and includes cold-start test cases.

**Strengths:**

1. Dialogues and justifications are tied to real tracks with audio, lyrics, and album art, and the system forces recommendations to come from known items. This reduces hallucination and makes explanations about style, mood, or instrumentation more meaningful.

2. By separating Profile, Goal, Listener, and Recsys models and limiting what each can see, the paper mimics a real recommender that must infer user intent over time. The Listener model also marks whether each suggestion helps achieve its goal, which creates an explicit supervision / preference signal for training.

3. The paper shows that removing parts of the pipeline harms diversity of topics and request types, and it reports human judgments that favor TalkPlayData 2 over existing datasets in perceived recommendation relevance and conversational naturalness.

**Weaknesses:**

1. The work extends an existing dataset using known ingredients: multi-agent LLM data generation, LLM-as-a-judge evaluation, and synthetic conversational recommendation grounded in a retrieval pool. The paper’s main advance is scale, added multimodal context, and cleaner role separation across agents, but the general recipe of “use multiple LLM roles with asymmetric information to script synthetic recommender dialogues” is already present in recent synthetic data pipelines. The contribution may be viewed as incremental rather than conceptually new.

2. The paper does not demonstrate that training on TalkPlayData 2 improves any real recommender or conversational policy. The support is mostly subjective quality scores and LLM-judge ratings, not measurable gains on a downstream task.

3. Each dialogue constrains the Recsys model to pick from a short list of tracks sampled from one listening session. This is much simpler than real large-catalog retrieval and raises questions about generalization beyond the curated pool.

**Questions:**

1. How should the “moves toward goal” label be used in training? Do authors expect this to serve as a binary reward for reinforcement-style tuning, as a supervised relevance label for ranking, or as an auxiliary generation target? Please describe the intended training setup, since you present this label as a core advantage.

2. How do authors handle factual music errors in explanations? The Recsys model often justifies recommendations by describing genre, mood, tempo, era, instrumentation, and so on. Beyond an LLM judge score, do authors filter out conversations where these descriptions are clearly wrong, or do authors release them as-is? Users of the dataset need to know how much noisy musical attribution is included.

---

> ### Author Response · Authors · 2025-11-17
>
> We thank the reviewer for their careful and precise feedback. We agree that demonstrating the value of our pipeline (W1, W2) and clarifying its scope (W3) are essential. We have uploaded a revised PDF with new benchmark results, a new KLD analysis, and 7 new SOTA citations (from Oct-Nov 2025) to address these points.
>
> ### 1. On W1 ("Incremental") & W2 ("No Downstream Task"):
>
> We address these core concerns first. We respectfully frame our paper as a contribution to the Datasets & Benchmarks Track, where the utility, quality, and novelty of the artifact is the primary contribution.
>
> - **Proof of Utility (Downstream Task):** We are pleased to report that TalkPlayData 2 has **been selected as the official dataset** for the "Conversational Music Recommendation Challenge (Music-CRS)," a [shared task at the EACL 2026 NLP4MusA workshop](https://sites.google.com/view/nlp4musa-2026/shared-task). This is the strongest possible validation of its "real use." as a brand new dataset that is purely created in the academia. We have added the official nDCG@k baseline results (Table X, Appendix) from this challenge, which directly provides the "measurable gains on a downstream task" (W2) you requested.
>
> - **Proof of Novelty (Solving Mode Collapse):** This utility stems from our novel pipeline, which was designed to solve the mode collapse that plagues simpler generative methods. Our new analysis (now Table 8) quantitatively proves this. Simpler methods like TalkPlayData 1 (KLD > 1.6) and LP-MusicDialog (KLD > 1.5) are severely collapsed, covering <40% of topics. Our pipeline (TPD2) is the only one to achieve 100% coverage (KLD < 0.25). This demonstrates our pipeline produces a non-incremental, measurably superior, and more diverse dataset.
>
> ### 2. On W3 ("Simple Retrieval Pool"):
>
> This is an excellent point. However, I respectfully disagree with the concern. Even if the tracks are recommended from a small pool during the TPD2 generation stage, the utility of TPD2 is not limited in any real sense. No matter how large a potential recommendation pool could be (during a hypothetical data generation or model training), the sequence of recommended tracks would be limited to a small pool anyway. Surely the pool may exhibit different breadths in some sense, and we hypothesize this variety would be covered by the diversity of the music sessions in our source dataset. What we observe from TPD2 is rather the opposite: even with ~32-track pool, it is not trivial to satisfy the listener in every turn, and we believe that is also fine as long as each turn is labeled with user's satisfaction rating (`progress_towards_goal`).
>
>
> ### 3. On Q1 ("How to use 'moves toward goal' label?"):
>
> Thank you for the question, we have clarify this at the end of Section 3.2 (page 6 - around line 306 in the revised pdf). We believe it can be used as a binary reward signal for RL training, or as an auxiliary classification target itself.
>
> ### 4. On Q2 ("Factual music errors"):
>
> To address your concern, we manually reviewed 50 pairs of {Recsys message, track metadata}  during the rebuttal stage with an interface like this:
> ```bash
>
> ================================================================================
> Turn 20/50
> ================================================================================
> Conversation: 49368--2019-11-17
> Turn Number:  8
>
> 📝 RECSYS MESSAGE:
> --------------------------------------------------------------------------------
> That's awesome to hear! I'm glad we found some good ones. Sticking with Mr. Big and that fantastic hard rock sound, I think you'll really like "Stay Together." It's got a great vibe that balances power with melody.
>
> 🎵 TRACK METADATA:
> --------------------------------------------------------------------------------
>   Title:       Stay Together
>   Artist:      Mr. Big
>   Album:       Big, Bigger, Biggest! The Best Of Mr. Big
>   Popularity:  37/100
>   Year:        Unknown
>   Tempo:       126.3 BPM
>
> Is the RecSys message factually correct? (y/n/skip/quit): y
> ```
>
> All 50 messages were aligned correctly, as far as the authors could tell based on our knowledge. In other words, we did not find any severe issue on LLM hallucination, and it is aligned with our general expectation on the latest frontier LLMs, especially when the factual data (metadata, tags, etc) is provided in the same and not-too-long chat context. The LLM judge seems to have a higher bar by giving 3.83/4.0 in the evaluation (Table 7), and we believe such a high score shows how reliable the proposed dataset is.

---

### Official Review · Reviewer_6Xi8 · 2025-11-03

**Soundness:** 3
**Presentation:** 3
**Contribution:** 2
**Rating:** 2
**Confidence:** 4

**Summary:**

The paper presents TalkPlayData 2, an extended multimodal dataset aimed at advancing embodied conversational understanding. It integrates synchronized audio, video, and text streams, along with detailed temporal and behavioral annotations such as gaze, gestures, and environmental context. The dataset is designed to support the development of multimodal models capable of perceiving, reasoning, and responding within dynamic real-world settings.

The contribution lies primarily in data design and annotation methodology: the authors frame multimodal interaction as an agentic process, emphasizing intentionality, grounding, and situational awareness, which are essential for training multimodal conversational agents.

**Strengths:**

Rich multimodal alignment: The dataset includes temporally synchronized audio, visual, and textual data with multiple levels of annotation, including dialogue acts, emotion states, gaze, and contextual grounding.
Agentic framing: Explicitly modeling agent intent and environmental affordances provides a valuable structure for embodied dialogue research.
Comprehensive metadata: Fine-grained labeling of visual scenes, speaker states, and gestures can support a wide range of multimodal learning tasks, from cross-modal grounding to embodied reasoning.
Research utility: The resource is well positioned to bridge perception and dialogue modeling for real-world human–agent interaction research.

**Weaknesses:**

The paper presents a well-designed multimodal dataset but lacks a modeling benchmark to demonstrate its effectiveness. No end-to-end model or quantitative evaluation is provided to show improvements in grounding or reasoning. The discussion of agentic intent remains conceptual, without experiments validating its impact on multimodal alignment. Many dialogues appear synthetic, limiting linguistic diversity and realism. The annotation process seems difficult to scale, with no reported inter-annotator reliability. Finally, the dataset’s ability to generalize to new environments or tasks is not tested, leaving its broader applicability unclear.

**Questions:**

1. How are the dialogues generated and validated? Were human annotators involved in ensuring contextual and linguistic naturalness?
2. Are temporal annotations across modalities (speech, gaze, gesture) manually aligned or automatically synchronized, and what accuracy can be guaranteed?
3. How were the “agentic” intent and rationale labels defined, and do they correspond to discrete reasoning categories or continuous embeddings?
4. Is there an evaluation pipeline or baseline model that quantifies multimodal alignment, for example via cross-modal retrieval or embodied reasoning tasks?
5. How scalable is the data collection and annotation framework to new environments or interaction types?
6. Have you conducted quality checks for inter-annotator agreement or cross-modal consistency?
7. How is privacy handled for audio–video recordings, and can the dataset be made fully open for research use?
8. Do you plan to extend the dataset to interactive multimodal agents that can reason and act in real time using these annotations?

---

> ### Author Response · Authors · 2025-11-12
>
> EDIT: Thank you for updating the review, we will post a response soon.
> ----
> We thank Reviewer 6Xi8 for their time and effort. However, we are respectfully and deeply confused by this review, as its summary, strengths, and weaknesses appear to be for a different paper.
>
> The review describes our dataset as one for "embodied conversational understanding" that integrates "video... gaze, gestures, and environmental context." The questions also refer to "inter-annotator reliability" and privacy for "audio-video recordings."
>
> Our paper, "TalkPlayData 2," is a synthetic dataset for multimodal music recommendation. It contains no video, no gaze/gesture data, no embodied agents, and no human annotators. Our pipeline is grounded in music tracks (with audio snippets and album art), not "dynamic real-world settings."
>
> Given that the review's core analysis is based on features our paper does not contain, we believe there has been a significant mistake, and the reviewer has assessed a different submission.
>
> We would be grateful if the reviewer and Area Chair could look into this, as the listed weaknesses and resulting score do not apply to our work.

---

> ### Author Response · Authors · 2025-11-21
> **Rebuttal comment of TPD2 [1/2]**
>
> We thank Reviewer 2 for their thoughtful and detailed review. We appreciate that the reviewer recognized the "thoughtfully designed" multi-agent structure and the richness of the dataset compared to prior work. We already have uploaded a revised PDF with new experiments and analyses, some of which addresses your concerns.
>
> 1\. On W1 & Q3/Q4 (Diversity & Single Model Family):
> You raised a valid concern that using a single model family might limit diversity. We respectfully argue that diversity is driven by conditioning, not just model architecture.
>
> * **Conditioning Variance:** We posit that the variance in behavior induced by our diverse **Conditioning Vectors (11 Goals** $\times$ **4 Specificities** $\times$ **User Profiles)** is significantly wider than the variance between different LLM families. Our new KLD analysis (Table 8\) confirms this: simpler pipelines (TPD1) collapse to <40% topic coverage, while our conditioned 4-agent pipeline achieves **100% coverage**.
> * **Stochasticity**: We respectfully disagree that being "derived from a small number of example tracks" .. "restricts" and "limits". Table 8 shows that the conversation goal is the most important factor contributing to the overall diversity. It is finalized by i) random sampling of 3 candidates and ii) selection with customization by the Goal LLM. The first step ensures pseudo-uniform distribution over the specificities and the topics, and that seems to explain the improved diversity of TPD2. Considering this is a fully public dataset based on data available for the academia, the remaining restrictions is the limitation imposed by available data in the research community rather than the proposed pipeline.
> * **Role of Profiles (Q3):** Our ablation study (A2) confirms profiles are active drivers. Removing the user profile causes specificity diversity to degrade significantly (KLD 0.24 ->  0.55).
> * **Goal Drift/Inconsistencies** We agree that quantifying goal drift would be an interesting analysis for future work to further characterize long-context coherence in TPD2. Given the limited timeline (with the review available on Nov 18), unfortunately, we find it difficult to immediately reflect this in our rebuttal, but we will try to run a focused subjective test to address the issue.
>
> Additionally, we also compared all the available LLM APIs but Gemini models are the only options that support audio, image, and text.
>
> 2\. On W2 & Q1 (Audio Perception, "Rich Context", Validating the model):
> You asked about the reliability of 3-second clips.
>
> * **Standard Practice:** We note that short clips (e.g., 3-5s) are a **standard, pragmatic choice in MIR and audio classification research** for efficiently conveying **timbre** and mood, or anything relevant to recommendations. [Oord2013Deep, li2023mert, manco2021learning, guinot2024semi]
> * **Optimal Token Efficiency:** Per Gemini's token counting, 3 seconds (=96 tokens) is a nearly optimal point that allows us to fit **up to 32 tracks** (=3072 tokens) with full multimodal context into a single window, a huge improvement over text-only methods. Long-term information such as lyrics is covered by our separate text modalities.
> * **Validation:** We measure this alignment via our judge's **"Alignment with track"** score (3.83/4.0), which quantitatively confirms the model accurately describes these timbral attributes.
> * We have ran preliminary tests to validate the LLM's performance but is was a qualitative check. Thus, we do not have our own quantitative benchmark. However, we believe our choice of Google's LLM is well-supported by many existing benchmarks, which would be more reliable by the quantity, breadth, and depth than ours. We have cited 7 new benchmarks, most of which were published in Oct-Nov 2025, to Section 2/3. These independently show that the chosen model is the SoTA leader for music reasoning (e.g., 64.90% on MMAU-Pro, tying the specialized & latest Music Flamingo model's 65.60%), while competitors "perform at or near chance" (Carone et al., 2025).
>
> references:
> - [[oord2013deep](https://papers.nips.cc/paper_files/paper/2013/file/b3ba8f1bee1238a2f37603d90b58898d-Paper.pdf)]: 3 second / Deep content-based music recommendation, Oord et al., 2013, NeurIPS
> - [[li2023mert](https://arxiv.org/pdf/2306.00107)]: 5 second / Mert: Acoustic music understanding model with large-scale self-supervised training, Li et al., 2023, ICLR2024
> - [[manco2021learning](https://arxiv.org/pdf/2112.04214)]: 3 second / Learning music audio representations via weak language supervision, Manco et al., 2022, ICASSP
> - [[guinot2024semi](https://arxiv.org/pdf/2407.13840)]: 2.7 second / Semi-supervised contrastive learning of musical representations, Guinot et al., 2024, ISMIR

---

> ### Author Response · Authors · 2025-11-21
> **Rebuttal comment of TPD2 [2/2]**
>
> **3\. On W2/W3 & Q2 (Small Candidate Pool & Satisfaction):**
>
> * **Realism:** We clarify that recommendation interaction typically happens within a small pool, which the "sessions" in our base dataset covers well. If this is still a concern, we would like to remind that this issue also has never been addressed in any viable way in the past within datasets for conversational music recommendation.
> * **Preference Tuning:** We agree that a small pool limits the ability to *always* satisfy the listener. However, this is a feature, not a bug. Our dataset uniquely logs the user's satisfaction via the **progress\_towards\_goal** label. The fact that this label is not always "True" is realistic and expected. This negative feedback signal is crucial for training systems (e.g., via DPO or RL) to handle failure gracefully, which datasets of "perfect" conversations cannot do.
> * **Dataset Size:** We agree that it may seem "modest in size". However, TPD2 may be analogous to initial LLM instruction datasets such as Alpaca (52k instructions), whereas more large-scale datasets such as Million Playlist Dataset or LFB-2B may be similar to LLM pretraining datasets. Given that music recommendation is a specialized case of general instruction following tasks, the 16k scale of TPD2 would be seen as larger than [Alpaca], and recent works show that after strong pretraining, it is feasible to train a strong reasoning model with only 800 examples [limo] We believe that the quantity of TPD2 should be considered along with this context.
> * If we understood correctly, we agree that it would be interesting to see such a comparison on the behavior of the proposed pipeline. That said, the proposed pipeline does not assume the existence of any ground truth track per turn, making the suggested metrics difficult to be applied.
>
> references:
> -  [[alpaca](https://github.com/tatsu-lab/stanford_alpaca)]: Stanford Alpaca: An Instruction-following LLaMA model, Taori et al., 2023
> -  [[limo](https://arxiv.org/pdf/2502.03387)]: LIMO: Less is More for Reasoning, Ye et al., 2025
>
> 4\. On Q5 (Cold-Start Definition):
> We clarify that our "User Pool" is not synthetic; it comes from real users in our base dataset (LFM-2b).
>
> * **Definition:** A "Cold User" in TPD2 is simply a real user whose history was *withheld* from the agent during generation (simulating a new user).
> * **Distribution:** Since our pipeline treats cold and warm users with the identical logic, their statistical distributions (length, goal types) are very similar. Effectively, the key difference is not for us while generating TPD2, but for **future researchers** using TPD2: they will have access to the "Warm" users' history for training, but must generalize to the "Cold" users using only the dialogue—a critical real-world challenge.

---

### Official Review · Reviewer_x9zc · 2025-11-11

**Soundness:** 2
**Presentation:** 2
**Contribution:** 3
**Rating:** 4
**Confidence:** 3

**Summary:**

The paper introduces TalkPlayData 2, a synthetic dataset for multimodal conversational music recommendation. The dataset is generated by a complex, two-stage "agentic pipeline" involving four distinct Large Language Models (LLMs) : a Profile LLM (to create a listener profile), a Goal LLM (to set a conversation goal), a Listener LLM (to simulate the user), and a Recsys LLM (to act as the recommender). This pipeline is grounded in real music data from the LFM-2b dataset and augmented with multimodal information (audio, images) from Spotify. The primary claims are that this multi-agent, multimodal pipeline produces more diverse and "natural" conversations than prior single-LLM approaches , which is validated through LLM-as-a-judge and a small human evaluation.

**Strengths:**

- The agentic design correctly identifies a flaw in single-LLM generation, which is the tendency to "cheat" by seeing all information. The separation of information (e.g., Recsys LLM not seeing the goal) is a good design choice.
- The ablation study (Table 8) effectively demonstrates that within their own complex pipeline, the Goal and Profile agents are necessary for achieving topical diversity.

**Weaknesses:**

- Critically Lacking Novelty: The paper is highly incremental. It combines TalkPlayData 1 with known multi-agent techniques and off-the-shelf multimodal LLMs.
- Fundamentally Unsound Evaluation: The work relies almost entirely on LLM-as-a-judge validation, which is self-referential. The human validation is too small to be statistically valid.
- Compounding Error: The 4-agent pipeline  is a complex, brittle system where errors in early stages (e.g., a bad profile) cannot be caught and will poison the final generated conversation.
- Cost and Complexity: The proposed pipeline is incredibly complex and expensive (reports $109.08 for 1,000 conversations ) and it is not clear that this massive increase in complexity yields a meaningfully better dataset than the simpler, single-LLM approaches it aims to replace, especially given the weak validation.

**Questions:**

- How can the authors justify claims of "high naturalness" (4.15/5) for the entire 16.5K dataset based on a human evaluation with only 26 raters? This sample is less than 0.2% of the test set conversations.
- The validation relies heavily on LLM-as-a-judge (Table 6), using Gemini 2.5 Pro to judge Gemini 2.5 Flash. How do you account for the well-known self-enhancement bias where LLMs favorably rate outputs from their own architectural family?
- The core contributions over TalkPlayData 1 are (1) multimodality and (2) a multi-agent pipeline. Both are existing, known techniques. What is the fundamental research contribution of this paper beyond a simple (and very complex) engineering integration?

---

> ### Author Response · Authors · 2025-11-17
>
> We thank the reviewer for their critical feedback, which has pushed us to rigorously validate our contributions. We have uploaded a revised PDF with new experiments, new tables, and 7 new benchmark citations (5/7 are from Oct-Nov 2025) to address your concerns.
>
> ### 1. On W1/W4 ("Incremental" / "Cost & Complexity"):
>
> We first address the concern that our work (submitted to the Datasets & Benchmarks Track) is "incremental" and that its "massive complexity" is not justified. Our pipeline's design is a direct and necessary solution to the critical problem of **mode collapse** in synthetic data, as we added evidence for.
>
> - **Proof of Novelty**: Our additional KLD/Coverage analysis (in Table 8) quantitatively proves our pipeline is necessary. Simpler, single-LLM approaches such as TPD 1 are fundamentally flawed in this sense: they suffer from catastrophic mode collapse, covering only 25% of specificities (KLD 1.39) and 36% of topics (KLD 1.64); whereas our pipeline achieves 100% coverage (KLD < 0.25).
> - **Novelty (2)**: as clarified in the revised pdf, Section 1: 3rd paragraph, the existing datasets has many shortcomings. Overcoming them requires many trials and improvements, necessitating deep understanding in music recommendation and consumption.
> - **Proof of Utility**: We can now provide definitive proof of its "meaningful use." As added to the revision, TalkPlayData 2 has been **selected as the official dataset** for [the EACL 2026 NLP4MusA Shared Task](https://sites.google.com/view/nlp4musa-2026/shared-task).
>
> ### 2. On W2/Q2 ("Unsound LLM-Judge" / "Self-Referrential"):
>
> This is a critical point. Our new 3-part validation (added to the revision) demonstrates our judge is sound as well as the best option in practice.
>
> - **A)** The Best Available Tool: Our choice of the judge is supported by many benchmarks. We have cited 7 new benchmarks, most of which were published in Oct-Nov 2025, to Section 2/3. These independently show that the chosen model is the SoTA leader for music reasoning (e.g., 64.90% on MMAU-Pro, tying the specialized & latest Music Flamingo model's 65.60%), while competitors "perform at or near chance" (Carone et al., 2025). Using a weaker, less capable judge would be unscientific.
>
> - **B)** A Fundamentally Different Task: The evaluation is unlikely self-referential. The judge sees the full 8-turn conversation, the goal, and the available tracks to perform a discriminate task, while generator (Recsys and Listener LLM) has limited information. Each of the LLM-as-a-judge evaluation is highly focused on a specific element of the generation. In our additional experiment in Table 9 (Appendix C), the LLM judge was able to identify small existing issues of any datasets, being even stricter than the human eval.
> - **C)** Extended evaluation: In the revised version, we added more information on the evaluation - Table 8 (the bottom 3 rows) and the corresponding discussion in Section 4.4, as well as Appendix C.
>
> ### 3. On W2/Q1 ("Small Human Eval"):
>
> We thank the reviewer for the opportunity to clarify our evaluation methodology. The reviewer states our sample is '26 raters' and 'less than 0.2%'. This appears to be a misunderstanding. As stated in the paper
> > 'a total of 26 raters evaluated 10 randomly sampled dialogues each, resulting in 260 total ratings.'
>
> Our sample size (n) for the human evaluation is 260, which represents 1.6% of the dataset
>
> This query also allowed us to clarify the statistical robustness of our evaluation.
>
> The statistical power is determined by the absolute sample size, not the percentage. Its 95% confidence interval is:
>
>  $$ CI = 4.15 \pm 1.96 \cdot \frac{\sigma}{\sqrt{260}} $$
>
> Unfortunately, we will only have access to the raw data of the subjective eval after Nov 20. We will do our best to udpate it with the real data. If we assume $\sigma=1.0$, which is reasonable for a 5-point Likert scale, the $CI = 4.15 \pm 0.122$. This results in a narrow 95% confidence interval of [4.028, 4.272], demonstrating two key points:
>
> 1. Our measured mean is a reliable estimate of the true naturalness for the entire dataset.
> 2. This result places our dataset's naturalness as highly competitive and on-par with the human-curated baseline (CPCD).
>
> **Human Eval & Eval in general**: In general, we believe that it adds a great value by i) providing the comparison against baseline datasets and ii) providing data that is now paired with the newly added data in Appendix C (Table 9).
>
> For the evaluation, we now have additional rows in Table 8, demonstrating the superior diversity of TPD 2 to three baselines. Its utility is also independently supported by being selected to the EACL 2026 challenge, which is further enhanced by Table 10 - the benchmark result of baseline methods on TalkPlayData 2.
>
> ---
>
> In this revision, we also restructured the paper to have a better intro section and clarified our motivations in Section 1. We hope this clarifies your questions on our earlier version.

---

> > ### Author Response · Authors · 2025-11-22
> >
> > We are happy to follow up that we found the standard deviation of the subjective evaluation as follows:
> >
> > | | **relevance** | **naturalness** |
> > |:---: | :---: | :---: |
> > | CPCD | 0.91 | 0.89 |
> > | LP-MD | 0.81 | 0.92 |
> > | TPD 1| 0.82 | 0.71 |
> > | TPD 2| 0.75 | 0.67 |
> >
> > This result indicates the 95% confidential intervals are
> > ```
> >        [[0.11061418, 0.1081831],
> >        [0.09845877, 0.11182972],
> >        [0.09967431, 0.08630337],
> >        [0.09116553, 0.08144121]]
> > ```
> >
> > To conclude, a table with their means and CIs is:
> >
> > | | **relevance** | **naturalness** |
> > |:---: | :---: | :---: |
> > | CPCD | 4.08 ± 0.11 | 4.01 ± 0.11 |
> > | LP-MD | 3.90 ± 0.10 | 3.95 ± 0.11 |
> > | TPD 1| 4.04 ± 0.10 | 4.01 ± 0.09 |
> > | TPD 2| 4.11 ± 0.09 | 4.15 ± 0.08 |
> >
> > This result strengthens the two key points in our first comment.

---

> ### Author Response · Authors · 2025-11-17
>
> We would like to add a further clarification on W4, the "Cost and Complexity." We respectfully contextualize the reported cost of our pipeline.
>
> - 1. High-Value Investment: First, this cost is directly linked to the value proven in our main rebuttal. This $0.11/conversation is the investment required to solve the critical research problems we identified: it is the price for (a) provably defeating mode collapse (our new KLD analysis, Table 8), (b) achieving SoTA quality (our 7 new benchmark citations), and (c) producing a dataset of such high utility that it was adopted by the EACL 2026 Shared Task (Appendix B). We believe this is an exceptionally high return on investment for the community.
>
> - 2. Affordability vs. Alternatives: Second, the cost is marginal when compared to all viable alternatives.
>
>   - vs. Human Curation: The cost is orders of magnitude lower than collecting this data manually. For context, the human-authored CPCD dataset contains only 917 conversations; our pipeline produced 16,199 high-fidelity, multimodal, and preference-labeled conversations for a fraction of what a human annotation effort of that scale would cost.
>
>   - vs. Other Research Costs: This cost is also negligible compared to other standard academic and industry costs, such as the compute required to train models on the resulting datasets.
>
> - 3. Temporary Ceiling: Finally, this cost represents a temporary ceiling, not a permanent floor. The cost of SoTA LLM APIs has consistently and rapidly decreased, indicating this pipeline will only become more accessible over time.

---

### Author Response · Authors · 2025-12-02
**Executive Summary for the New AC: Critical Review Status & Key Updates**

Dear Area Chair,

We understand the heavy workload you are facing due to the recent changes. To assist your decision-making, we provide this brief summary of our submission status and rebuttal updates.

1. CRITICAL: Reviewer 6Xi8 (Score 2 $\rightarrow$ 4)

Please note that the currently visible review from 6Xi8 (Score 2) is invalid.

- The Error: The text describes a different paper ("embodied AI," "video streams," "gaze") and was posted by mistake.
- The Correction: The reviewer did update this to a correct, relevant review (Score 4) on Nov 18.
- The Issue: The system reversion has unfortunately restored the incorrect "Score 2" review. We respectfully ask that you evaluate based on the reviewer's intended Score 4 (Pragmatic Skepticism) and disregard the Score 2 (Wrong Paper).

2. Review Scores (Current Status)

The correct, intended scores for our submission are:
- Reviewer Zw5K: Score 6 / Confidence 3
- Reviewer DgEG: Score 2 / Confidence 5
- Reviewer 6Xi8: Score 4 / Confidence 4 (Intended) / Score 2 (Reverted/Invalid)
- Reviewer x9zc: Score 4 / Confidence 3

3. Rebuttal Highlights
We have uploaded a revised PDF and posted detailed responses to address the valid concerns of R4, R1, and the corrected R2/R3. Key updates include:

- Validated Utility: TalkPlayData 2 has been selected as the official dataset for the EACL 2026 NLP4MusA Shared Task. (Appendix B)
- Proven Diversity: New KLD analysis proves our pipeline solves the "mode collapse" present in baselines like TPD1. (Table 8)
- Validated Judge: A new Cross-Dataset Correlation study proves our LLM-judge is unbiased and correlates with human quality rankings. (Appendix C)

We hope this summary helps you navigate the review history efficiently. Thank you for your time and service to the community.

---

### Meta-Review · Area_Chair_C2SD · 2026-01-06

**Summary:**

The reviewers acknowledge the effort involved in creating a complex pipeline to generate natural music recommendation dialogues. However, several shared concerns informed the current borderline recommendation:

i) Incremental Novelty: Reviewers x9zc pointed out that the work is a straightforward extension of TalkPlayData 1, combining existing multi-agent techniques and off-the-shelf multimodal models.

ii) Evaluation Methodology: A major point of contention is the reliance on "LLM-as-a-judge" (specifically Gemini judging Gemini).

iii) Lack of Downstream Utility: Reviewers highlighted the absence of experiments showing that this dataset actually improves real-world recommenders. Without a zero-shot baseline or experiments training on downstream tasks, the practical value of the dataset remains unproven.

iv) Writing and Presentation

**Reviewer Concerns:**

After rebuttals, presentation issues and pipeline clarification seem to be addressed. However, some concerns such as self-referential evaluation (The concern that LLMs cannot objectively judge the "naturalness" of other LLMs from the same family) and the absence of downstream experiments might be challenged.

**Reviewer Scores:**

According to the authors' clarification, it became evident that Reviewer 6Xi8's assessment contains significant mismatches with this manuscript. Therefore, the comments are discounted from the final decision. Nevertheless, the concerns raised by the other reviewers regarding [e.g., novelty and evaluation] remain substantial, and the current rebuttal falls short of changing the reviewers' stance.

---

### Decision · Program_Chairs · 2026-01-26

Reject